# Maximising Affordability of Real-Time Colorimetric LAMP Assays

**DOI:** 10.3390/mi14112101

**Published:** 2023-11-15

**Authors:** Simon Strachan, Moutoshi Chakraborty, Mohamed Sallam, Shamsul A. Bhuiyan, Rebecca Ford, Nam-Trung Nguyen

**Affiliations:** 1School of Environment and Science, Griffith University, Nathan Campus, Brisbane, QLD 4111, Australia; moutoshi.chakraborty@griffithuni.edu.au (M.C.); mohamed.sallam@griffithuni.edu.au (M.S.); rebecca.ford@griffith.edu.au (R.F.); 2Queensland Micro- and Nanotechnology Centre (QMNC), Griffith University, Nathan Campus, Brisbane, QLD 4111, Australia; sbhuiyan@sugarresearch.com.au (S.A.B.); nam-trung.nguyen@griffith.edu.au (N.-T.N.); 3Centre for Planetary Health and Food Security, Griffith University, Nathan Campus, Brisbane, QLD 4111, Australia; 4Griffith Institute for Drug Discovery, Griffith University, Nathan Campus, Brisbane, QLD 4111, Australia; 5Sugar Research Australia, Woodford, QLD 4514, Australia

**Keywords:** portable device, on-site diagnostics, precision agriculture, LAMP

## Abstract

Molecular diagnostics have become indispensable in healthcare, agriculture, and environmental monitoring. This diagnostic form can offer rapid and precise identification of pathogens and biomarkers. However, traditional laboratory-based molecular testing methods can be expensive and require specialised training, limiting their accessibility in resource-limited settings and on-site applications. To overcome these challenges, this study proposes an innovative approach to reducing costs and complexity in portable colorimetric loop-mediated isothermal amplification (LAMP) devices. The research evaluates different resistive heating systems to create an energy-efficient, cost-effective, and compact device to heat a polydimethylsiloxane (PDMS) block for precise temperature control during LAMP reactions. By combining this novel heating system with an off-the-shelf red-green-blue (RGB) sensor to detect and quantify colour changes, the integrated system can accurately detect *Leifsonia xyli* subsp. *xyli*, the bacteria responsible for ratoon stunting disease (RSD) in sugarcane. The experimental validation of this system demonstrates its ability to detect the target pathogen in real time, making it an important development for low cost, portable, and easy-to-use molecular diagnostics in healthcare, agriculture, and environmental monitoring applications.

## 1. Introduction

Portable diagnostics have become increasingly important in various fields, including clinical [1,2,3], phytopathology [4], and forensic science [5,6]. Molecular diagnostics, which can detect deoxyribonucleic acid (DNA) signatures, are highly valued and often considered the gold standard for diagnostics [7]. This technology allows for accurately detecting specific DNA markers in diverse settings. One such molecular technique, loop-mediated isothermal amplification (LAMP) [8], has gained significant attention due to its simplicity, speed, and versatility [9]. Compared to its robust and highly accurate counterpart, polymerase chain reaction (PCR), LAMP offers significant advantages for on-site testing scenarios, where immediate results are crucial for informed decision making [10]. Although PCR has been introduced into the field [11,12], this approach requires precise temperature control via a thermocycler, whereas the isothermal nature of LAMP provides affordability and simplicity in equipment.

Traditionally, colorimetric LAMP uses indicators like phenol red or HNB to create a visual colour shift [13,14]. The resulting colour change helps identify a positive result quickly, making it ideal for on-site molecular testing, especially in resource-limited settings. By incorporating real-time colorimetric detection during the amplification cycle, the method provides valuable semi-quantitative data, facilitating quick decision making in situations where traditional binary results may not be enough. Recent advances, including image analysis [15], pH monitors [16], and smartphones [17], have significantly improved the technique’s quantification capabilities. Smartphone-based image analysis offers a portable solution for clinical and plant pathogen diagnostics at the point of decision, although additional instrumentation is required [17,18]. In addition, pH microelectrodes provide a rapid and reliable target diagnosis within minutes, serving as an alternative to real-time, portable quantification [16].

The proposed system combines LAMP’s isothermal capabilities with a heating element to initiate reactions and a red-green-blue (RGB) sensor to detect colour changes. Choosing the right heating source is crucial for reliable and efficient performance [10]. Isothermal methods, such as LAMP, significantly simplify the heating process, allowing for the development of unique heating systems that are energy efficient and more suitable for on-field applications [19]. Electricity-free chemical heating, like exothermic heating [20] or using energetic compounds to undergo phase transformation at the desired incubation temperature [21], displays instrumentation-less amplification that is low power and does not require trained personnel to operate. However, techniques such as this require manufacturing complexities and do not have the desirable availability and market readiness. That is why resistive heating elements have been employed for portable amplification processes, most commonly to heat aluminium blocks [22]. Velders et al. [23] found that a PDMS mould with an embedded nichrome coil and thermistor provided a highly selective heating unit. The low thermal conductivity of PDMS alleviates the need for thermal insulation, making it a suitable choice for a cost-effective, easily manufactured device.

This paper studies the optimal heating source for a PDMS mould to be included within a portable LAMP device based on time to reach the desired temperature, power consumption, cost, manufacturability, and ease of access. Evaluating heating sources based on these criteria can identify the most suitable option for on-site applications that balances performance, affordability, and practicality. The detection accuracy is as crucial as the amplification sensitivity, so the heating element must accommodate colorimetric detection. PDMS allows for this, with its transparency being a highly desirable feature for optical detection and, in turn, mitigating the need for complex housing to accommodate the real-time analysis that would be required if a traditional aluminium block were to be introduced.

The current colour detection methods for LAMP are often expensive and require complicated instruments [16,24], which makes them difficult to access. Therefore, the aim was to create an affordable and accessible real-time colour detection system for on-site LAMP testing. Papadakis et al. [15] have coined the term quantitative colorimetric LAMP (qcLAMP) to describe the use of digital image analysis with a Raspberry Pi computer and camera for colour changes during LAMP amplification. However, by alleviating the necessity for high processing power in the Raspberry Pi and a camera, an affordable RGB sensor with a similar algorithm for quantitative analysis can accomplish the same task while being more cost-effective and readily available. This approach has been successfully used in other areas of research, such as plant monitoring [25], biomass monitoring [26], and food and beverage quality assurance [27,28,29]. Therefore, adapting this method for real-time colorimetric analysis in LAMP offers a practical solution that overcomes the cost and accessibility barriers of traditional approaches, making on-site LAMP testing more widely accessible.

This paper presents the design, analysis, and performance evaluation of a real-time colorimetric detection system, potentially contributing to the advancement of on-site LAMP-based testing applications. The results from this study will pave the way for the broader adoption of cost effective and efficient heating sources, facilitating the practical implementation of the proposed system and enhancing its impact on on-site molecular testing.

## 2. Materials and Methods

### 2.1. Materials

The TCS34725 RGB sensor and TCA9548a I2C multiplexer were purchased from Adafruit (New York, NY, USA). Also, 30 × 40 mm and 45 × 100 mm film heaters were purchased from Amazon (Bellevue, WA, USA). The 33 W/4 A Peltier modules were purchased from Jaycar Electronics (Sydney, NSW, Australia). Heater cartridges were purchased from Bilby3D (Clayton, VIC, Australia). All other electronics were purchased from Jaycar Electronics. The SYLGARD silicone elastomer 184 and a curing agent were purchased from Dow Corning Corporation (Midland, MI, USA). All PCBs were designed on EasyEDA by JLCPCB (Hong Kong, China) and developed by the same company. The software layer of the device was designed on the Arduino IDE (Ivrea, Italy). All components were designed using Autodesk Fusion 360 (San Francisco, CA, USA) and then printed with filament from Prusa Research (Prague, Czech Republic) on a Prusa i3 MK3S+ printer. WarmStart^®^ Colorimetric LAMP from New England Biolabs (Ipswich, MA, USA). The open-access LAMP primer design software Primer Explorer V5 was used to design the six target-pathogen-specific primers. All oligonucleotide sequences were purchased from Integrated DNA Technologies (Coralville, IA, USA).

### 2.2. Experimental Setup for Heating Block Evaluation

A custom mould (10 × 23 × 47 mm^3^) with a PCR tube as a template was used to create the PDMS microreactor, ensuring consistent contact with subsequent tubes for optimal heat transfer. The block was designed to heat four reactions, including positive and negative controls and two unknown targets. A poly(methyl methacrylate) (PMMA) mould was created by laser cutting the walls and base, with holes for the PCR tubes. This PDMS mould was used for all heating sources except the heating cartridges that required an additional step. The PDMS was prepared by mixing a 10:1 silicone-to-curing agent ratio and vacuuming to remove air bubbles. After being poured into the mould, the PDMS was incubated for 2 h at 75 °C.

Four heating elements were tested for optimised heating of the PDMS block for efficient and accurate real-time colorimetric detection: a 24 V 30 W film heater, a 5 V 1 W film heater, a 15 V 33 W Peltier device, and the 12 V 40 W heater cartridge embedded within the PDMS block. The same block of PDMS was used for testing the film heaters and Peltier to ensure an unbiased comparative analysis. However, a specifically manufactured PDMS block was required for testing the heater cartridges. The PDMS mould was fabricated by embedding the cartridges below the PCR tubes for direct and efficient heating. These cartridges were placed on a 3D-printed side wall with a hole to hold them. Figure A1 shows the mould configuration for setting the heating cartridges within the PDMS.

An experimental setup was designed to determine the optimal heating element for the PDMS block and the time to reach the desired temperature required for effective LAMP reactions. The film heaters and Peltiers were positioned on both sides of the mould, directly contacting the PDMS, to ensure maximum surface area and maintain a consistent heat rate, facilitating effective heat transfer and temperature distribution, as shown in Figure 1a. While the heater cartridges were embedded within the PDMS.

Figure 1a shows the custom housing designed and 3D printed using polyethylene terephthalate glycol (PETG) to hold each heating element. PETG was selected because of its high thermal resistance and because it is an easy-to-print filament. The housing was designed to fit various heating element configurations, ensuring consistent temperature measurements. Negative temperature coefficient (NTC) thermistors were placed at the bottom of each PCR tube hole in the PDMS block to monitor the temperature (Figure 1b). The thermistors accurately measured temperature by detecting changes in resistance that were processed by a microcontroller. The thermistors were secured to the bottom of the tube holes with Stars 922 heat-conductive adhesive and threaded through the holes of a lid to ensure consistent contact (Figure A2a).

An Arduino Uno microcontroller was used to interface with the thermistors and heating elements to control the temperature and manage the experiment timing. This allowed for automated temperature monitoring and data collection. A 30 V, 5A bench power supply provided the voltage and current to the heating elements and allowed for controlling their performance. Integrating the NTC thermistors, Arduino Uno microcontroller, and bench-top power supply created a controlled environment for measuring and monitoring the reaction temperature (Figure A2b). Our setup enabled gathering accurate temperature data and evaluating the capabilities and stability of the heating elements.

Measurements were taken to evaluate the performance of each heating system, placed under their optimal heating voltages, from the point of thermal equilibrium until a temperature increase of 30 °C. The systems were set up with their heating elements connected in parallel to ensure they operated at maximum voltage and received the necessary power for optimal heating performance. Monitoring the temperature increase over time gathered crucial data on each system’s heating efficiency at their maximum heating capabilities. The thermistors secured at the bottom of the tube holes continuously monitored the temperature at these critical locations, ensuring that the thermal profile remained consistent and uniform throughout the experiment. This data facilitated the comparison of heating systems and enabled the identification of the most efficient heating source for the PDMS block based on its ability to reach and maintain the desired temperature.

### 2.3. RGB Sensor Calibration

Precise and accurate colour detection is crucial in the proposed real-time colorimetric detection system to determine the presence or absence of target DNA accurately and sensitively. To achieve this, sensor calibration compensates for variations in sensor readings, environmental factors, and potential inaccuracies in colour perception. However, limited lighting conditions within the enclosure can lead to variations in colour perception and inaccurate readings. Interference from ambient light, inconsistent light sources, and inadequate illumination intensities can negatively impact the output of the RGB sensor. Therefore, sensor calibration is necessary to overcome these challenges and ensure accurate colour detection, even under suboptimal lighting conditions.

The ColorChecker (X-Rite, Grand Rapids, MI, USA) colour chart/reference chart was used as the benchmark for calibration. The ColorChecker is used in academic research, such as spectral imaging, due to its meticulous design, consistency, and detailed spectrophotometric measurements [30]. Colours ranging from natural to grayscale produce measurements in standard RGB (sRGB), the colour space used to represent and display images on electronic screens with illuminant D65, and a standardised light source used to represent daylight with a colour temperature of 6500 Kelvin (K). Along with International Commission on Illumination (CIE) L*a*b* values, which is the colour space that was defined by the CIE to represent colours in a way that is more perceptually uniform to the human eye, an illuminant D50 2-degree observer, 2-degree denoting how the average human observer with normal colour vision would perceive colour if looking straight at it. However, each ColorChecker measurement patch requires conversion into the CIE XYZ colour space. This serves as a foundational colour space that provides a device-independent representation of colours and enables the colours to be mathematically related to human visual perceptions of colours. Finally, a chromatic adaptation was applied to obtain the correct LED illuminant, LED-B5, to compare the results from the RGB sensor within the enclosure. The algorithm used to produce the correction matrix can be found in Appendix A. Figure 2 illustrates the algorithm in flow chart format to better display the conversion into colour spaces and the equations used.

To take measurements of the ColorChecker, the RGB sensor was set up in an environment to simulate the light of the enclosure with the following conditions: (i) constant distance between the light source, object, and sensor; (ii) constant light reflectance (beam angle); and (iii) constant software conditions (i.e., gain and integration time). A 3D-printed housing was designed to have a fixed angle and keep a consistent distance of 10 mm between the sensor and the sample. The same material and colour were chosen to make the housing as close to the full system enclosure as possible: PETG and black (Figure A3). The average reading of each colour patch’s raw 16-bit data was recorded and stored.

Once each patch’s 16-bit raw data had been converted with a correction matrix to CIE XYZ LED-B5, a comparison using CIE1976 ${\Delta E}_{ab}^{*}$, which used a Euclidean formula to acquire a qualitative means of determining the colour difference between two colours, was used. Consequently, a 59.8% improvement was seen when applying the correction matrix. More importantly, the colours yellow and magenta displayed impressive improvements from 65.6 to 4.20 and 34.7 to 8.83, respectively (considering that $\;{\Delta E}_{ab}^{*}$ = 2.00 is an unnoticeable difference to the human eye and 0 is entirely no difference). This is illustrated in Table A1 with 8-bit sRGB colours used for visual inspection (it should be noted that these sRGB colours are not accurate and were used for visual comparison of the differences only).

The correction matrix improved the real-time colorimetric detection system by calibrating the TCS34725 RGB sensor. This was especially important due to the limited lighting conditions within the full system enclosure. The calibration process fine-tuned the sensor’s response and sensitivity to accurately detect and distinguish colour changes, specifically in the yellow and magenta ranges, which are essential for the proposed qcLAMP protocol that employs phenol red [15]. The calibration procedure addressed the limitations of constrained illumination, which mitigated variations in ambient light interference and inconsistent light sources. As a result, the system is more accurate and reliable in detecting colours.

### 2.4. Colour Analysis and Performance Evaluation of RGB Sensors

The phenol red in the LAMP mix is pH-sensitive and responds to increased protons by the DNA polymerase. As a result, the higher the amount of target nucleic acid in the sample, the more significant the colour change, which the RGB sensor can detect. The microcontroller then interprets and plots the real-time curve difference between green and blue values, which is referred to as the colour index value (CIV). This was previously tested and demonstrated by Papadakis et al. [15] as a robust way of quantifying and graphing the colour change between magenta and yellow. A positive result is confirmed when the CIV exhibits three consecutive measures of positive slope and exceeds five times the baseline noise levels.

To account for any potential light absorption and scattering effects caused by the PDMS heatblock. A comparative analysis of various wavelengths relevant to the experimental setup (magenta and yellow) with and without PDMS was undertaken to determine whether PDMS would affect results. It was concluded that the PDMS shifted the results but overall had an insignificant impact on the outcome due to CIV normalising to zero at the beginning of each experiment. To further ensure that there were no imperfections in the PDMS, it was ensured that the block was free of blemishes and that any bubbles that formed during the setting process were eliminated by using a vacuum. Moreover, the experiments were conducted under consistent illumination conditions. This involved carefully regulating the intensity and angle of the light to minimise the effects of light scattering. By using uniform illumination parameters across all experiments, it was possible to reduce potential discrepancies caused by light scattering. This helped to minimise any interference with the light source.

### 2.5. System Design and Assembly for Preliminary Testing

The optimised heating block and calibrated RGB sensor were combined to create a unified system before testing and evaluating the system’s performance. A schematic representation of the system assembly is shown in Figure 3 to easily illustrate the design fundamentals used. Integrating the heating element and RGB sensor concurrently enables temperature control and real-time colour detection.

The system was designed using a modular approach, integrating multiple components to achieve its functionality. The primary components are the microreactor and sensing unit, which work in tandem to enable precise temperature control and analysis during molecular testing (Figure A4). The system’s main body was fabricated using 3D printing to ensure the stability and cohesion of all components. The holding assembly was printed from polylactic acid (PLA) due to its ease of use, cost-effectiveness, and rapid prototyping capabilities. On the other hand, the microreactor holder, which held the PDMS mould, was printed from PETG due to its superior durability, making it an ideal choice for housing the delicate PDMS microreactor. Additionally, PETG exhibits excellent temperature resistance, enhancing its suitability for this critical system component.

A primary Atmega-328p microcontroller board, an Arduino, managed the electronic system. This microcontroller enabled the execution of various processors and algorithms that were critical to the system’s functionality and the data acquisition process. The microcontroller outputs necessary information, such as time, temperature, RGB/XYZ values, and whether the results are positive, to a file that can be analysed later. To achieve precise temperature control, a PID algorithm was implemented. The Arduino also facilitated essential tasks such as converting raw RGB signals to XYZ data, controlling LEDs, and driving the MOSFET on the main PCB (Figure A5). To accommodate the integration of four TCS34725 RGB sensors with fixed I^2^C addresses, the I^2^C multiplexer TCA9548A was employed. Additionally, two separate and purpose-designed PCBs were the ideal mounting platforms for the RGB sensors, enabling connection to the multiplexer.

### 2.6. LAMP Method for Testing the Combined System

The performance of the system was evaluated with the bacteria *Leifsonia xyli* subsp. *xyli* (*Lxx*), a significant pathogen that causes ratoon stunting disease (RSD), which has a detrimental impact on sugarcane crops [31]. Primers were designed targeting a 210 bp conserved section of the intergenic spacer (IGS) region between 16S and 23S rRNA genes (GenBank accession no. AE016822.1) [32], corresponding to positions 135,389–135,598 in the *Leifsonia xyli* subsp. *xyli* strain CTCB07 complete genome. For LAMP analysis, two loop primers (*Lxx*LF and *Lxx*LP), two outer primers (*Lxx*F3 and *Lxx*B3), and two inner primers (*Lxx*FIP and *Lxx*BIP) were used (Table A2).

The bacterium, *Lxx,* was obtained from sugarcane xylem sap collected at the Sugar Research Australia (SRA) Woodford Research Station, Woodford, Queensland. The sap samples were extracted from sugarcane stalks infected with ratoon stunting disease (RSD), which were obtained from an RSD screening trial. The sap was collected from two sugarcane varieties, SRA22 and SRA26, which were selected because they are resistant and susceptible, respectively. To obtain the sap, the stalks were cut at the base using sterilised secateurs and scrubbed with ethanol, and gloves were used to eliminate any extraneous material. Positive pressure was then applied to the stalk from the basal nodal and internodal regions using a small air compressor with a soft rubber cup attached. Approximately 2 mL of xylem sap was extracted for each of the varieties and collected into 15 mL centrifuge tubes. The samples were immediately kept at 4 °C during transportation to the laboratory for further study. To simulate on-field testing, a simple heat lysis step of 95 °C was used for DNA extraction [33,34,35].

The LAMP mixture was prepared according to the manufacturer’s instructions with some modifications. The WarrmStart^®^ LAMP reactions were carried out in a 25 μL mixture containing 2.5 μL of a 10× LAMP primer concentration, 12.5 μL of a 2× WarmStart^®^ Colorimetric LAMP Master Mix, 2 μL of DNA template, and 8 μL of nuclease-free water. An additional drop of mineral oil was added to mitigate evaporation. The mixture was incubated at 65 °C for 30 min in the combined system or in a BioRad T100 thermal cycler. To confirm the accuracy of the integrated system, the same LAMP reaction was conducted in parallel using a T100. The end-point images obtained with the integrated system were compared with end-point images from the thermal cycler, which established a reliable benchmark. Furthermore, the experiment included a no-template control (NTC), which allowed for the detection of false positives and ensured that the system’s specificity in detecting the target was not interfered with by non-target elements. The products were then stored at 4 °C before further analysis. The colorimetric LAMP products were evaluated visually, and samples that turned yellow were considered *Lxx*-positive, while those that remained pink were considered *Lxx*-negative, according to the manufacturer’s instructions. Nuclease-free water served as no-target controls (NTC). Each assay was performed in triplicate.

## 3. Results and Discussion

### 3.1. Evaluation of Heating Options

The heating characteristics of the designed systems were assessed and provided valuable insights into their performance and suitability. These insights served as the basis for informed decision making in developing an on-site molecular testing device (Figure 4).

The 12 V, 40 W heater cartridge within the PDMS block makes for a more compact system, minimising the need for external interfaces and additional wires. This feature makes the system easy to assemble and improves its manufacturability. The heater cartridge also provided uniform and efficient heating, which resulted in reliable and reproducible outcomes in the colorimetric detection process.

Among the heating systems evaluated, the 5 V film heater demonstrated a compact size that could be used in situations that prioritise energy efficiency and lower power consumption. However, this film heater exhibited a limited heating capacity, resulting in only a 10.71 ± 1 °C temperature increase after 10 min. On the other hand, the 24 V film heater produced significantly faster heating, reaching a 30 °C increase in just 2 min and 8 s. This film heater was initially trialled due to its high power capacity and compatibility with the PDMS block design. With 30 W, the film heater heated up quickly, reducing the time needed for efficient LAMP reactions. It also distributed heat evenly across the PDMS block, ensuring reliable LAMP results. Both film heaters were cost-effective and easily accessible, costing as little as AUD 3 or may be designed and purchased in a short timeframe if a specific size is needed. However, the 5 V film heater is less suitable for applications that require rapid and efficient molecular testing due to its limited power and slower temperature increase. While the 24 V film heater can heat up quicker, it is not suitable for integration due to heat containment problems, which is a significant drawback and would require additional insulation. Therefore, both film heaters require additional manufacturing and design costs that would need to be implemented to produce adequate results for the device.

The Peltier system displayed even faster heating than the 24 V film heater, achieving an average 30 °C increase in 1 min and 44 s. However, its high power requirement of 15 V and 4 A and the insufficient cooling capabilities for this power range posed limitations to practical implementation as this would require high costs for desirable power. Moreover, its cost and manufacturability are more limited compared to film heaters. Peltier modules are only available in given sizes and cost more than AUD 100 for custom designs. Although Peltier devices can be specifically manufactured, they are more complex than film heaters and therefore increase the cost of the device.

In contrast, despite requiring additional preparatory work and fabrication of the PDMS, the embedded heating cartridge demonstrated remarkable performance by achieving a 30 °C increase in only 57 s. Moreover, the use of PDMS not only transferred the heat but also acted as a thermal capacitor that effectively contained the heat, minimising the impact on the external ambient temperature while heating the sample tubes. With the heater cartridge within the PDMS block, it made for a more compact system, minimising the need for external interfaces and additional wires. This feature made the system easy to assemble and improved its manufacturability. The heater cartridge also provided uniform and efficient heating, which resulted in reliable and reproducible outcomes in the colorimetric detection process. Heater cartridges are reasonably priced, as they are commonly used in 3D printer hot ends, and their sizes, while not adjustable, are sufficient for use in the PDMS mould.

The outcomes of this study had important implications for selecting the components and materials for the construction of the portable device. Overall, these findings enabled informed decision making in selecting the optimal heating system, aligning with the requirements for rapid response, power consumption, cost effectiveness, manufacturability, and ease of access for developing reliable on-site molecular testing devices.

### 3.2. Optimising the Heater Cartridge

Next, an experiment was undertaken to determine the suitable power consumption ratings for the heater cartridges, which were selected as the optimal heating element for an on-site molecular testing device. The experiment involved varying the current control to 2 A, 3 A, and 4 A. At 2 A, the heater cartridges took approximately 2 min and 44 s to achieve a 30 °C temperature increase. However, increasing the current to 3 A reduced the heating time to 1 min and 27 s. At 4 A, the desired temperature increase was reached in just 57 s (Figure 5). Therefore, to determine the adequate power required from a battery pack, the switching circuit, seen in Figure A5, was introduced to determine the current draw from 7.4 V. This voltage was chosen to simulate two series-linked 3.7 V, 2600 m-Ah lithium polymer batteries. With a current draw of approximately 2.5 A, holding a consistent temperature of 65 °C for 30 min with a PID algorithm was found to be 233 mAh. Assuming a battery management system is incorporated, an expected 44 samples could be analysed before recharging is required. Thus, the battery pack was determined to provide sufficient power to achieve the desired temperature increase within a reasonable timeframe. This power configuration ensured efficient and effective heating for the on-site molecular testing device, enabling reliable and timely delivery of results. Moreover, the data recorded from the external thermistors provided evidence of the even temperature distribution within the reaction setup, which confirmed the reliability of the experimental conditions.

### 3.3. Combined System Results

During the LAMP reaction, dynamic changes in colour occurred, which allowed for the amplification to be visually monitored in real time by using CIV. This is shown by the colorimetric curves that indicate the amplification of a synthetic sequence for the target *Lxx* and *Lxx*-infected sap (as seen in Figure 6). The test was deemed positive at 17 min 10 s ± 51 s, 21 min 21 s ± 16 s, and 33 min 21 s ± 1 min 19 s for 10^7^, 10^5^, and 10^1^, respectively. The known infected sugarcane sap was also determined to be positive, with SRA26 being deemed positive at 20 min and 56 s ±1 min 2 s, and SRA22 at 34 min and 37 s ± 2 min 16 s. This provides preliminary data, showcasing differing time-to-positive responses to different bacterial loads between the susceptible and resistant varieties.

The qcLAMP reactions conducted using the integrated system have significant implications for on-farm diagnostics. When accompanying the semi-quantitative analysis with the naked-eye evaluation, the user’s end goal of determining whether an infection is present or not can be supplemented with the real-time analysis. This offers numerous advantages, particularly in agricultural settings where rapid and accurate disease detection is crucial for effective disease management and crop protection. With further testing required to better understand the correlation between time-to-positive and bacterial loading, a calibration curve can be derived to provide additional quantitative results. Therefore, it enables more timely decision making on disease control measures. Users can more confidently implement targeted interventions to reduce the risk of disease spread and optimise crop health with the rapid and accurate detection of target pathogens on-site.

There are various avenues to be explored to boost the device’s potential and user-friendliness. Firstly, encasing the device and having tests run off battery power will allow for proper testing on-site. Further optimisation of the heating element and temperature control system can also be pursued to achieve even faster reaction times while maintaining precise temperature control. Additionally, integrating connectivity options such as wireless data transfer and cloud-based analytics can streamline data management and facilitate remote monitoring and analysis. Furthermore, collaboration with agricultural experts and stakeholders can provide valuable insights into specific disease challenges and user requirements, guiding the device’s continuous improvement and customization to meet the needs of different on-site settings.

Implementing qcLAMP reactions into affordable devices holds great promise for on-site diagnostics. With the inclusion of the single-step 95 °C heat lysis step for sample preparation, it could significantly enhance the potential of any LAMP device, making it an ideal choice for point-of-need applications. Continued research and development efforts are required, as this study shows proof-of-concept for the detection of *Lxx* using a synthetic template and preliminary testing on infected sap. Further testing is needed on bacterial culture and xylem saps. This will enhance the device’s performance and expand its applications, making it a valuable tool for agricultural disease management.

The diagram illustrated in Figure 3 has been converted into a user-friendly, portable device. The experimental arrangement illustrated in Figure A4 has been fully enclosed. An Arduino hat has been utilised to interact with the various integrated circuits, while two lithium batteries are responsible for powering the device; this is illustrated in Figure A6a. The device’s physical appearance is displayed in Figure A6b. The entire system is priced at an affordable AUD 48.55 (Table A3). This cost-effective design not only ensures accessibility for various research and field applications but also significantly reduces the financial barriers associated with advanced molecular testing techniques. Therefore, this approach offers a practical and economical solution for molecular testing.

## 4. Conclusions

This research presents a novel integrated system that combines a microreactor with a high-precision RGB sensor for real-time colorimetric LAMP reactions. The device shows great promise in on-site diagnostic testing, directly offering rapid and accurate pathogen detection in agricultural settings. By streamlining the diagnostic process and reducing costs and complexities, this innovative device can adjust the way disease management is conducted and empower farmers with timely information. The successful integration of the microreactor and RGB sensor sets the stage for further advancements in on-farm diagnostics and paves the way for broader applications in diverse fields. Continued research and development efforts will undoubtedly enhance the device’s capabilities, making it a valuable tool for addressing pressing challenges in molecular diagnostics and beyond. Overall, this study sheds light on the potential of portable qcLAMP systems and highlights their relevance in molecular diagnostics.

## Figures and Tables

**Figure 1 micromachines-14-02101-f001:**
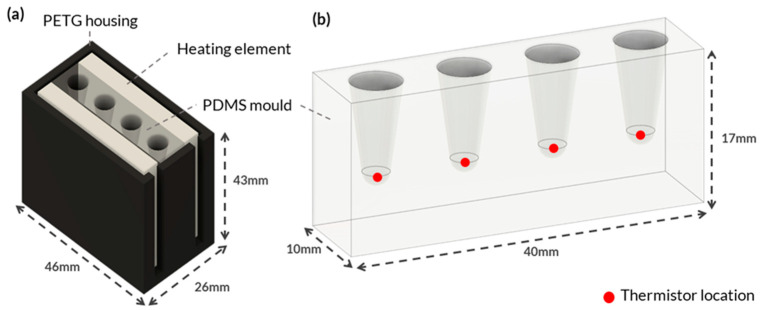
Integrated System Configuration for Heating Element and PDMS Mould. This figure showcases an integrated system configuration that comprises two main components: (**a**) 3D-printed PETG housing engineered to hold the various heating elements securely and to create a thermal sandwich with the PDMS material. (**b**) The PDMS mould was specifically designed to accommodate multiple PCR tubes for on-site molecular testing. The red dots in the figure represent the locations of NTC thermistors affixed to the PDMS.

**Figure 2 micromachines-14-02101-f002:**
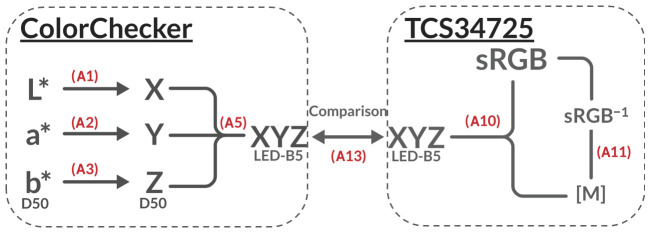
Algorithm flow chart: The flow chart presents the algorithm for converting colour spaces, illustrating the step-by-step process to have comparable benchmark from the ColorChecker and TCS34725 RGB sensor data. The chart is split into two segments. Left for the ColorChecker and right for the TCS34725. The flow chart highlights the equations used for each step of the colour space conversion, with the implemented equations in Appendix A denoted in red.

**Figure 3 micromachines-14-02101-f003:**
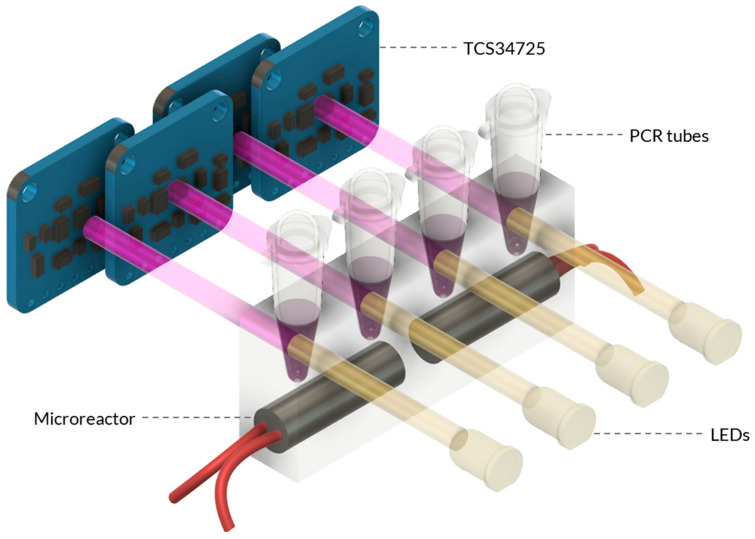
Schematic of the Combined Microreactor and Detection System. The combined system consists of the microreactor and the colour sensing unit. The microreactor is designed with heater cartridges that are embedded within the PDMS material. Adjacent to the microreactor, the sensing unit is positioned, featuring the TCS34725 RGB sensor assembly. The sensor assembly is strategically placed to detect and quantify the colour changes in the heated samples within the microreactor.

**Figure 4 micromachines-14-02101-f004:**
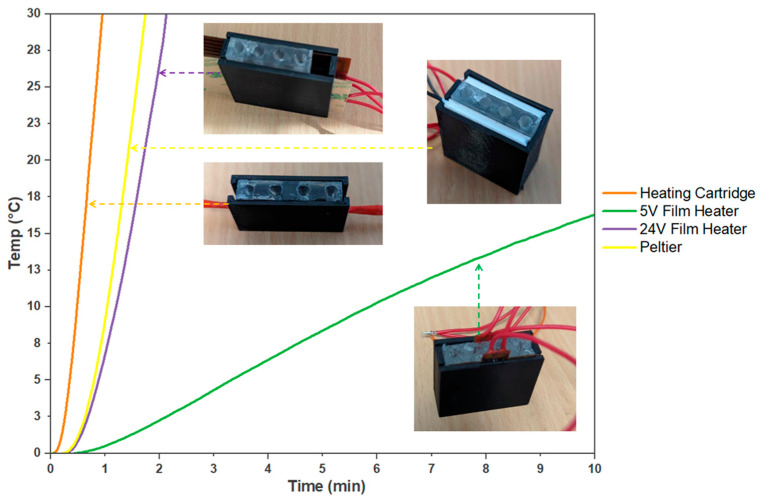
Comparison of Heating Systems for On-site Molecular Testing. The evaluated heating systems are a 5 V film heater, a 24 V film heater, a Peltier system, and an embedded heater cartridge. The curves indicate the time required for each heating system to achieve a 30 °C temperature increase (*n* = 3).

**Figure 5 micromachines-14-02101-f005:**
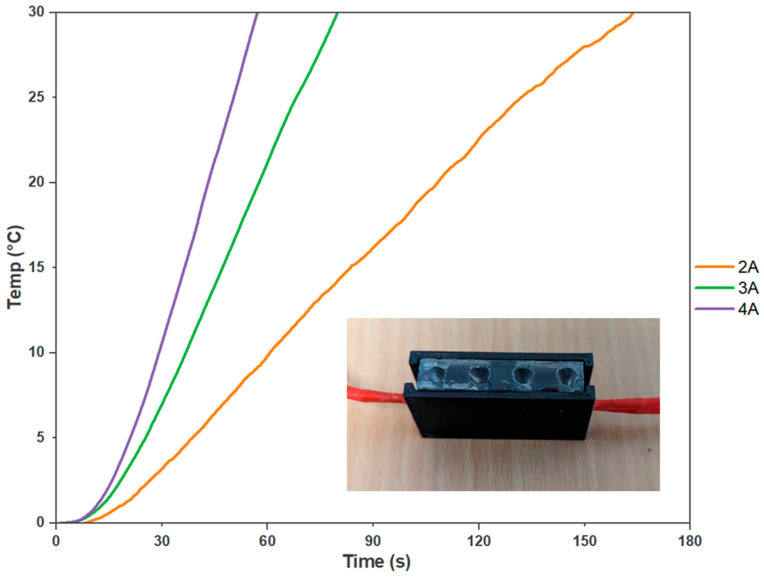
Heater Cartridges Power Consumption Analysis. The curves depict the relationship between power consumption and the heating performance of the on-site molecular testing device’s cartridges. The results demonstrate that higher current ratings lead to faster heating times, with 2 A taking 2 min and 44 s, 3 A requiring 1 min and 27 s, and 4 A achieving the desired temperature increase in just 57 s (*n* = 3).

**Figure 6 micromachines-14-02101-f006:**
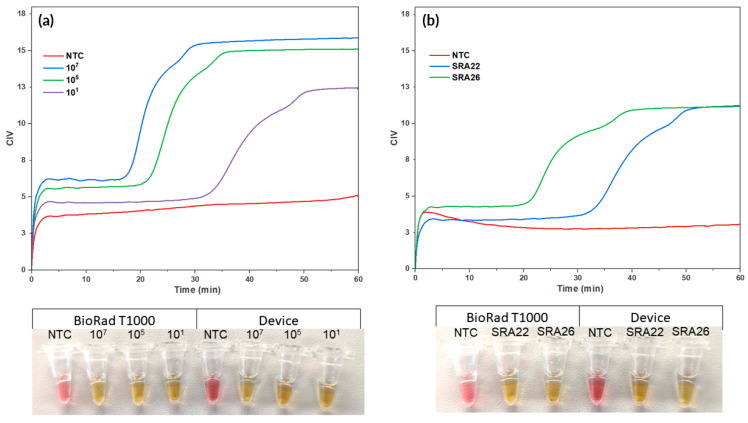
Real-time Colorimetric LAMP Reactions using the Combined Assembly. Real-time colorimetric LAMP reactions were performed using the integrated system. The colorimetric curves illustrate the real-time amplification of (**a**) a synthetic sequence for the target *Lxx* and (**b**) *Lxx*-infected sap. A positive result, indicating successful amplification of the target *Lxx*, was determined at 17 min and 10 s ± 51 s, 21 min 21 s ±16 s, and 33 min 21 s ±1 min 19 s for 10^7^, 10^5^, and 10^1^, respectively. The sugarcane sap was also determined positive, with SRA26 deemed positive at 20 min 56 s ±1 min 2 s and SRA22 at 34 min 37 s ±2 min 16 s. Picture of end-point reactions of NTC and synthetic copies of *Lxx* in device and in T100 (**left**) and end-point reactions of NTC, SRA26, and SRA22 in device and in T100 (**right**).

## Data Availability

Data are contained within the article.

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
