# Peer review of "Maximising Affordability of Real-Time Colorimetric LAMP Assays"

_micromachines, 2023, doi:10.3390/mi14112101_

Round 1

Reviewer 1 Report

Comments and Suggestions for Authors

This paper proposes a structured solution for a real-time colorimetric LAMP assay, combining a microreactor with RGB sensor for real-time color change detection. However, I have several doubts/issues about the general novelty of this work and about the fact that the study is a preliminary proof-of-concept work. The overall reliability of the proposed system should be confirmed by relevant and real applications. For example, the system was tested with 10^7 target copies, a too high concentration value that is quite far from real cases. Indeed, a key point in the development of such systems is the evaluation of the LAMP efficiency at the lowest target concentrations (10^2, 10, 1 target copies), near the reaction detection limit, to verify the reliability of the system, including the temperature control. Moreover, the paper hypothesizes the use of the device for semi-quantitative applications, without the support of preliminary data, which could thus truly justify the use of an instrumental color shift readout rather than a naked-eye one.

Comments on the Quality of English Language

N/A

Reviewer 2 Report

Comments and Suggestions for Authors

The authors designed a PDMS-based heating component and combined it with an RGB sensor to detect color changes during LAMP reactions. Overall, the device design is interesting from an easy to use  point of view. I recommend accepting this paper after minor revisions upon addressing the following questions:

1. Since the low cost for on-site application is the main focus of this work, it would be impressive for the authors to provide an estimated cost or price for the total materials used in the device.

2. The authors should clarify how they confirmed that the temperature field and distribution were uniform in the reaction tubes.

3. Although PDMS is transparent, it can still absorb light at different wavelengths or cause light scattering. The authors should specify whether they considered the influence of blank PDMS and its interaction with light during the detection process.

Round 2

Reviewer 1 Report

Comments and Suggestions for Authors

The authors have well addressed my points.

Comments on the Quality of English Language

can be slightly improved